# Chromosome-level assembly of the water buffalo genome surpasses human and goat genomes in sequence contiguity

Wai Yee Low [1], Rick Tearle[1], Derek M. Bickhart [2], Benjamin D. Rosen [3], Sarah B. Kingan [4], Thomas Swale[5], Françoise Thibaud-Nissen[6], Terence D. Murphy [6], Rachel Young [7], Lucas Lefevre [7], David A. Hume[8], Andrew Collins[9], Paolo Ajmone-Marsan [10], Timothy P.L. Smith[11] & John L. Williams [1]

Rapid innovation in sequencing technologies and improvement in assembly algorithms have enabled the creation of highly contiguous mammalian genomes. Here we report a chromosome-level assembly of the water buffalo (*Bubalus bubalis*) genome using single-molecule sequencing and chromatin conformation capture data. PacBio Sequel reads, with a mean length of 11.5 kb, helped to resolve repetitive elements and generate sequence contiguity. All five *B. bubalis* sub-metacentric chromosomes were correctly scaffolded with centromeres spanned. Although the index animal was partly inbred, 58% of the genome was haplotype-phased by FALCON-Unzip. This new reference genome improves the contig N50 of the previous short-read based buffalo assembly more than a thousand-fold and contains only 383 gaps. It surpasses the human and goat references in sequence contiguity and facilitates the annotation of hard to assemble gene clusters such as the major histocompatibility complex (MHC).

[1] The Davies Research Centre, School of Animal and Veterinary Sciences, University of Adelaide, Roseworthy, SA 5371, Australia. [2] Cell Wall Biology and Utilization Laboratory, ARS USDA, Madison, WI 53706, USA. [3] Animal Genomics and Improvement Laboratory, ARS USDA, Beltsville, MD 20705, USA. [4] Pacific Biosciences, Menlo Park, CA 94025, USA. [5] Dovetail Genomics, Santa Cruz, CA 95060, USA. [6] National Center for Biotechnology Information, National Library of Medicine, National Institutes of Health, Bethesda, MD 20894, USA. [7] The Roslin Institute and Royal (Dick) School of Veterinary Studies, University of Edinburgh, Easter Bush, Midlothian, United Kingdom. [8] Mater Research Institute–University of Queensland, Translational Research Institute, Woolloongabba, Brisbane, Queensland 4102, Australia. [9] Genetic Epidemiology and Bioinformatics, University of Southampton, Southampton SO16 6YD, UK. [10] Department of Animal Science, Food and Technology – DIANA, and Nutrigenomics and Proteomics Research Center – PRONUTRIGEN, Università Cattolica del Sacro Cuore, Via Emilia Parmense 84, 29122 Piacenza, PC, Italy. [11] US Meat Animal Research Center, ARS USDA, Clay Center, Ne 68933, USA. Correspondence and requests for materials should be addressed to J.L.W. (email: john.williams01@adelaide.edu.au)

A finished, accurate haplotype-resolved reference genome is necessary to understand the biology of a species, manage genetic diversity and, in the case of livestock, to apply genomic selection for genetic improvement[1]. However, despite advances in sequencing technologies, our ability to generate long contiguous DNA sequence reads is still limited, necessitating the use of a number of assembly algorithms and technologies to piece together the genomic jigsaw. For smaller haploid genomes, such as bacteria, complete assembly is now possible at relatively low cost[2] but the same does not apply to larger complex diploid or polyploid genomes. Mammalian genomes contain large families of repeats that are difficult to span, even with longer sequence reads, which, together with insufficient sequence coverage, result in breaks in sequence contiguity. Therefore, additional data types are required to correctly order and orient contigs. Fully assembling a mammalian genome is still challenging, and even the current human genome assembly (GRCh38), that has received considerable input of money and resources from more than 10 institutions and over 1000 researchers, still contains hundreds of gaps[3].

The latest PacBio single-molecule sequencing technologies[4] deliver mean read lengths above 10 kb, with reads as long as 60 kb[5]. This has facilitated the high quality assembly of mammalian genomes, including the gorilla[6] and the goat[7]. However, the relatively low throughput and higher error rates (~11–15%) remain a problem. Fortunately, PacBio sequencing errors appear randomly distributed, therefore, with sufficient depth, a consensus with high per base sequence quality can be achieved. Besides PacBio, other long-read sequencing platforms such as Oxford Nanopore are being used to assemble genomes at high accuracy[8].

Even with the improvement in long-read sequencing, additional approaches are required to accurately scaffold contigs. Hi-C[9], a modified version of chromosome conformation capture (3C)[10], identifies in vivo chromatin interactions across the whole genome, with the majority of interactions occurring within the same DNA molecule, often over many hundreds of kb. Chicago[11], a modified form of Hi-C, uses chromatin reconstituted in vitro with interactions limited to ~100 kb. The combination of Chicago followed by Hi-C enables contigs to be ordered and orientated at short- and long-range, respectively. Using both, the scaffolding processes create large scaffolds reaching to full length chromosomes.

Collapsing haplotypes from diploid organisms in genome assemblies can lead to errors in the sequence resulting from differences between homologous chromosomes[12]. One solution is to sequence haploid clones, as demonstrated by the use of tiled fosmids to assemble the human genome[13]. However, this approach requires the generation of clones, which is technically difficult and may introduce errors (e.g. chimeric clones). Complete haplotype-resolved diploid assembly has now been demonstrated using parental genotype data to separate sequence into haplotypes prior to assembly[14]. However, the ultimate goal would be to phase haplotypes from a single organism without having to generate clones or sequence the parents. The release of FALCON-Unzip[15] and more recently, FALCON-Phase[16] provides an advance towards this goal. FALCON-Unzip takes advantage of long reads to generate haplotigs (i.e. a contig consisting of a sequence with sufficient variation to define an alternative haplotype). FALCON-Phase combines PacBio and Hi-C data to resolve phase between haplotigs, thereby creating longer phased regions.

Here we present a near-finished genome assembly for the water buffalo (*B. bubalis*), a mammal with 25 chromosomes and a genome size of 2.66 Gb, which is comparable to human. The genome assembly was created using PacBio long reads assembled using FALCON-Unzip and scaffolded with Chicago- and Hi-C-based chromatin interaction maps. Illumina paired-end sequence was used for indel correction. This assembly strategy for the *B. bubalis* has achieved high sequence contiguity and accuracy, facilitating a substantially improved gene annotation and providing an exceptionally high-quality reference genome sequence for a species with global economic relevance.

## Results

**De novo assembly of a *B. bubalis* genome.** A female Mediterranean buffalo with the same bull as the paternal and maternal grandsire was used for sequencing. Sequence data comprised: ~75x PacBio Sequel long-reads, ~24x Chicago reads, ~58x Hi-C reads, and ~82x Illumina paired-end reads. The diploid FALCON-Unzip[15] assembler produced an initial PacBio-based contig assembly with 953 primary contigs, N50 of 18.8 Mb and a total length of 2.65 Gb (Fig. 1, Table 1). The assembler also generated a combined 1.53 Gb of haplotype-resolved sequence, or 58% of the total length of the primary contigs. The alternate haplotype sequence from the unzipped regions was output as 7956 haplotigs[16,17]. The haplotig N50 was 0.394 Mb and the longest haplotig was 2.77 Mb. Only the primary contigs were used in downstream scaffolding but the resolution of haplotypes improved contiguity and the accuracy of the assembly[12,14].

Scaffolding of the primary contigs was carried out in a series of HiRise analyses, initially using the Chicago data, followed by inclusion of the Hi-C reads. The HiRise program checks for incorrectly assembled contigs and introduces breaks, some of which were incorrect. The contig breaks were therefore classified as: (1) a break introduced into a region with the expected PacBio coverage, (2) a break in a region with an unusually high PacBio coverage, and (3) a break in a region of unusually low PacBio coverage (Supplementary Figure 1). A HiRise break in the first category was considered a false break. In total, 69/108 HiRise Chicago breaks and 4/6 HiRise Hi-C breaks were classified as false breaks and ignored. The most likely explanation for the high count of false breaks is where there is phase shift in the assembly between haplotigs (Supplementary Figure 2). This serial scaffolding step produced 509 scaffolds with an N50 of 117.2 Mb.

To further improve the assembly, sequence continuity was assessed by generating linkage disequilibrium (LD) maps for each of the 457 contig joins in the major 29 scaffolds that represent the 25 buffalo chromosomes. LD was assessed based upon SNP genotypes of 529 animals obtained using the current 90 K buffalo Axiom chip (see Methods). A total of 119 contig joins were found to be associated with LD jumps and also interrupted conservation of synteny with the cattle or goat sequence. These were considered potential mis-assembly points and were manually inspected, resulting in 18 scaffolds being reordered (Supplementary Note 1). Three pairs of scaffolds were joined to maintain LD, one on each of chromosomes 12, 21, and 25. The LD guided corrections produced longer scaffolds, which conserved synteny with the cattle and the goat genomes.

The final assembly, UOA_WB_1, after gap filling and error correction, covered the 25 buffalo chromosomes with only ~1% bases in 484 small unplaced scaffolds. All buffalo chromosomes were scaffolded in an order consistent with the buffalo whole genome radiation hybrid (RH) map and conserved synteny with the homologous *Bos taurus* (UMD3.1) chromosomes[18] (Fig. 2). As the RH data were not used to order or orient the scaffolds, this provides-independent evidence that the contig assembly and scaffolding are accurate. Additionally, the chromosome sizes and proportion of sequences aligned to corresponding homologous *B. taurus* chromosomes are in good agreement (Supplementary Figure 3 and Supplementary Table 1). It is noteworthy that, for all

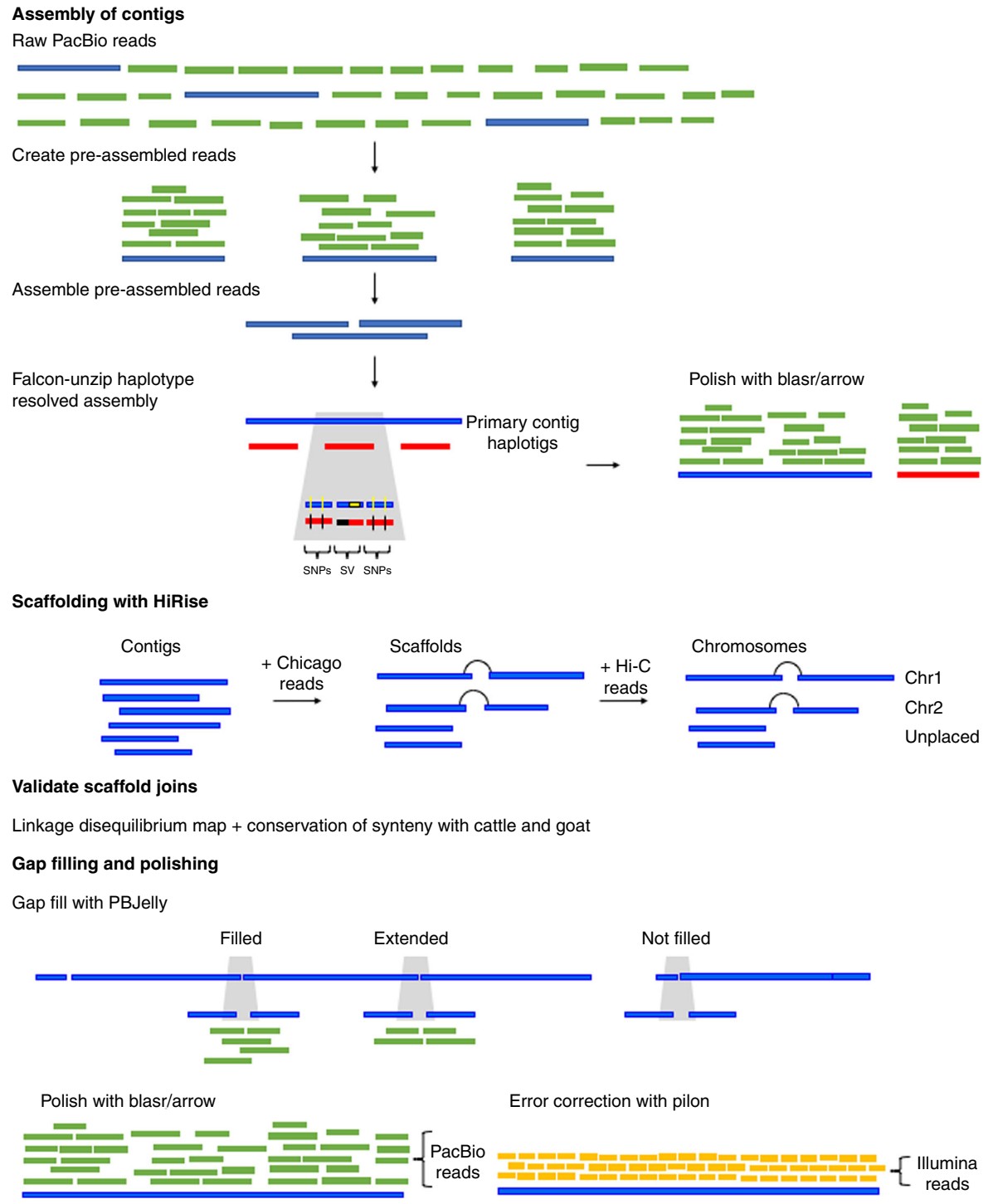

**Fig. 1** An overview of assembly methods. Contig assembly was carried out with the diploid assembler FALCON-Unzip to produce primary contigs and haplotigs. It began with selection of longest "seed" reads and shorter reads were aligned to them to create pre-assemble reads using a consensus approach. The primary contigs were carried forward to the scaffolding step that began with Chicago reads for short range scaffolding (1–100 kb) with HiRise. Then long-range scaffolding (10–10,000 kb) was carried out with Hi-C reads to cluster scaffolds to the chromosome level. Each join of contigs to create a scaffold was checked against an LD map and for conservation of synteny with cattle and goat. Then long-reads were used to fill gaps and polish the sequence, followed by indel correction with short reads

five of the sub-metacentric buffalo autosomes, the scaffolds span the centromeres.

**Assembly benchmarking**. The previous de novo water buffalo assembly (UMD_CASPUR_WB_2.0) was generated mainly from Illumina paired-end reads[19], which were assembled with MaSuRCA[20]. The resulting genome was highly fragmented, with the final assembly containing 2.84 Gb scattered in 366,983 scaffolds with a contig N50 of ~22 kb. Both UOA_WB_1 and UMD_CASPUR_WB_2.0 assemblies were benchmarked with the same assembly evaluation pipeline used to validate other long-

**Table 1 Assembly statistics**

| Assembly | Software | Assembly level | Number of sequences[a] | Number of gaps | N50 (Mb) | Assembly size (Gb) |
|---|---|---|---|---|---|---|
| PacBio | FALCON-Unzip | contig | 953 | 0 | 18.8 | 2.654 |
| PacBio + Chicago | HiRise | scaffold | 737 | 255 | 30.3 | 2.654 |
| PacBio + Chicago + Hi-C | HiRise | scaffold | 506 | 488 | 117.2 | 2.654 |
| UOA_WB_1 | PBJelly, Aarow, Pilon | chromosome | 25 | 383 | 117.2 | 2.622 |

[a]There are 484 unplaced contigs in the final chromosome-level assembly. These unplaced contigs comprise ~1% of total bases in the assembly and are not counted in the final UOA_WB_1 assembly size

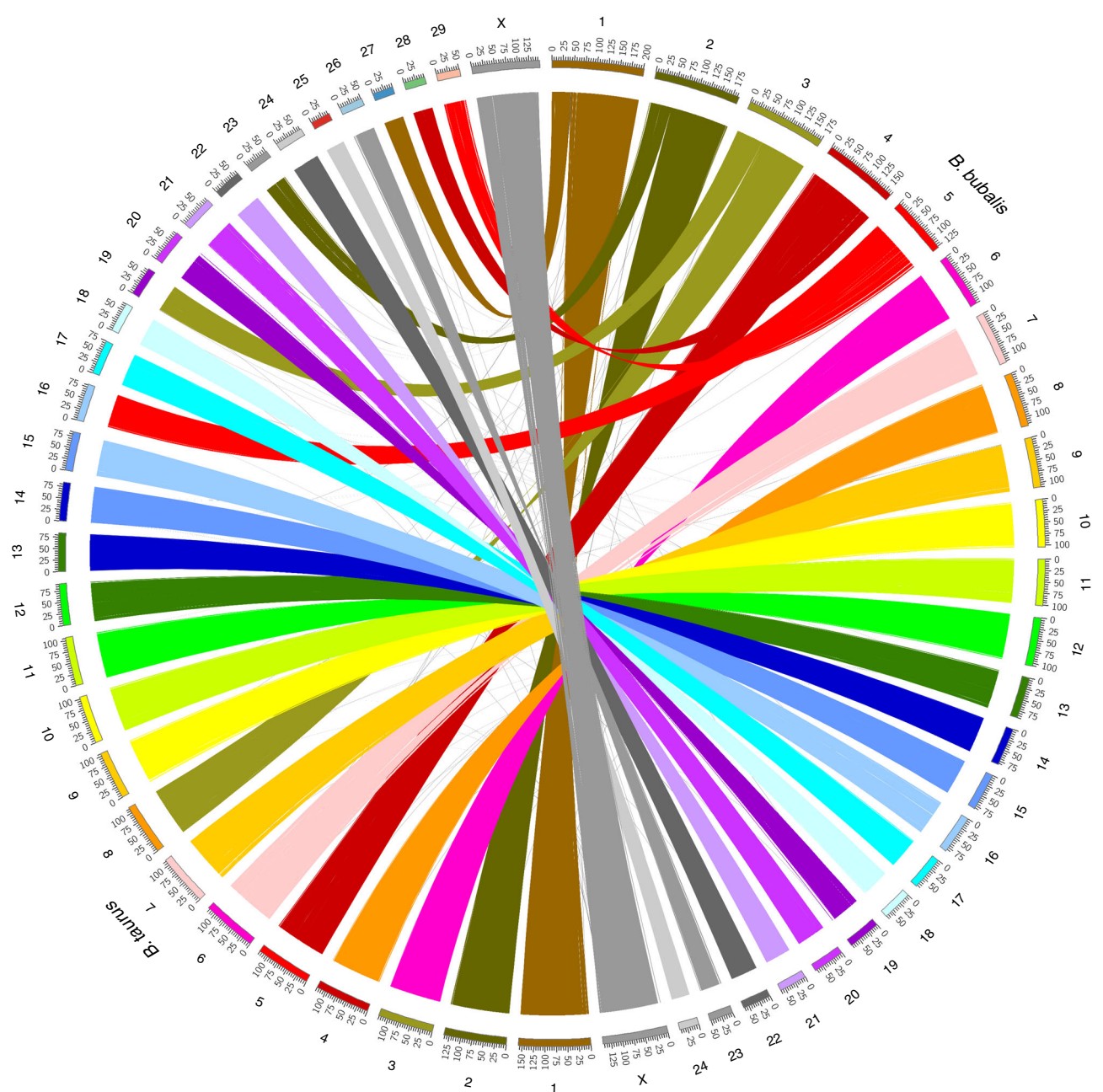

**Fig. 2** A circos plot of *B. bubalis* chromosome mapping to *B. taurus*. Chromosome 1–5 in *B. bubalis* are sub-metacentric and clear mapping to the expected homologous *B. taurus* (UMD3.1) chromosomes is found. Conservation of synteny of all *B. bubalis* chromosomes to *B. taurus* matched the whole-genome RH map

read reference assemblies[7] (Supplementary Note 1 and Supplementary Table 2). The per-base substitution quality values (QVs) for the UMD_CASPUR_WB_2.0 and for the UOA_WB_1 reference assemblies were 36.46 and 41.96, respectively. As the QV represents the phred-scaled probability of an incorrect base substitution in the assembly, a difference of 5 QV points indicates that UOA_WB_1 contains nearly half an order of magnitude fewer single nucleotide errors than UMD_CASPUR_WB_2.0. The contig N50 and scaffold N50 in UOA_WB_1 have a 1023-fold and 83-fold improvement, respectively, over the previous short-read based assembly (Supplementary Table 3).

Contigs constructed from long-reads should, in principle, be better than those produced from short-reads, as the former will span longer repeat regions. However, it is rare to be able to directly compare long-read to short-read-based assemblies of a complex genome with all sequencing data from the same individual. Both UMD_CASPUR_WB_2.0 and UOA_WB_1 were produced from the same female water buffalo, Olimpia. Contigs from UMD_CASPUR_WB_2.0 were aligned using nucmer[21] to the new UOA_WB_1 assembly to assess the larger structural

differences (50–10,000 bp) using Assemblytics[22]. The UOA_WB_1 assembly reported here is partly phased and has a genome size of 2.66 Gb; whereas, the short-read buffalo assembly (UMD_CASPUR_WB_2.0) is a mosaic of haplotypes and was highly fragmented, with the 2.84 Gb of assembled sequence included in 366,983 scaffolds with a contig N50 of ~22 kb. The fragmentation and inclusion of a mosaic of haplotypes in the short-read assembly in part explains the larger size. Therefore, differences between the two assemblies may arise from heterozygous alleles rather than true difference with UOA_WB_1. To test this, the haplotigs that represented 58% of the genome, were aligned to UOA_WB_1. A total of 12.5% of the structural differences called from the short-read assembly matched with the haplotigs (Fig. 3a). However, 9170 structural differences that comprise 3.3 Mb are likely to be assembly errors in UMD_CAS-PUR_WB_2.0; the majority being missing sequence (Fig. 3b, c). A total of 19 regions each larger than 8 kb, were missing from the previous assembly. Although Olimpia has one common grand-sire, and therefore a substantial amount of inbreeding, the level of heterozygosity was sufficient to assemble haplotigs which

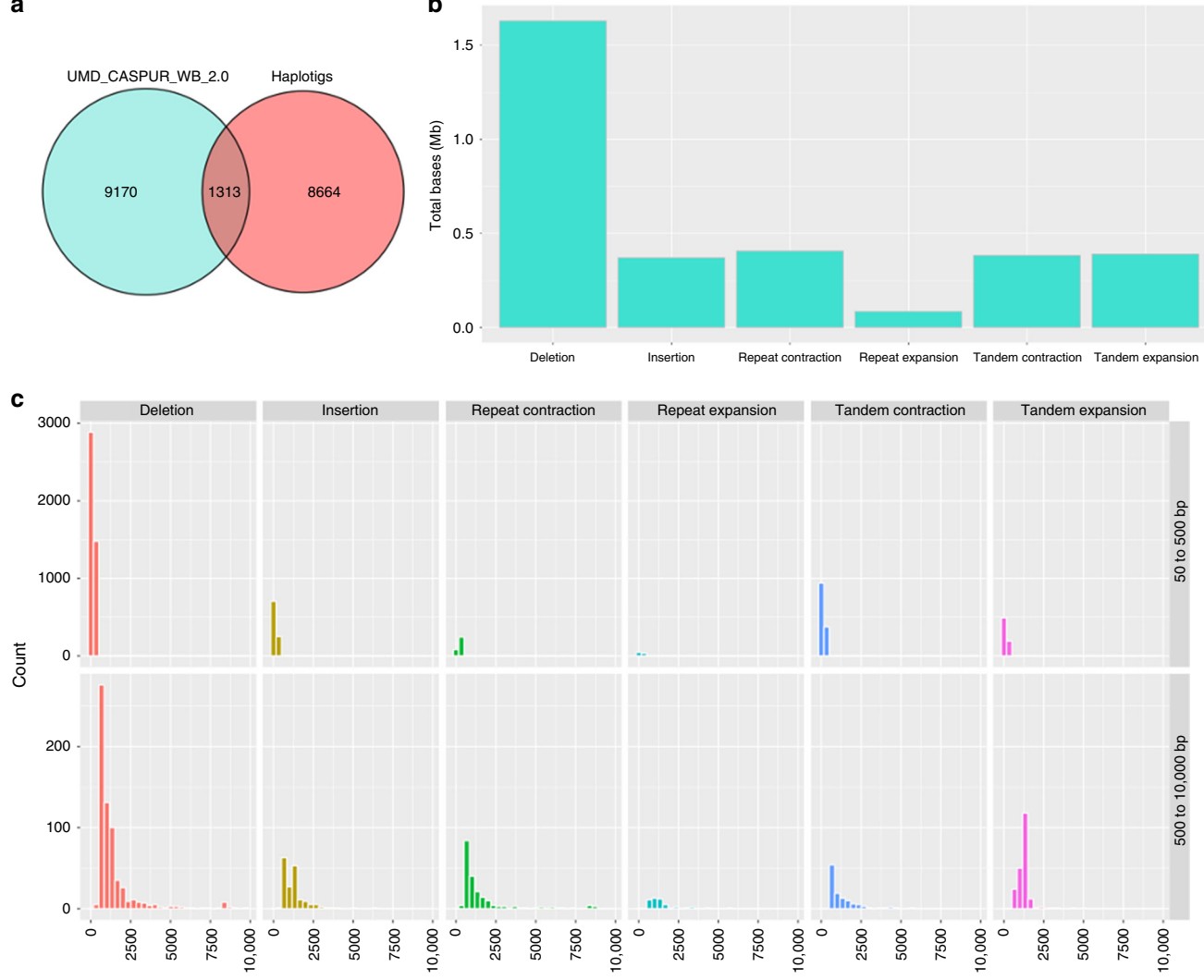

**Fig. 3** Structural differences between UMD_CASPUR_WB_2.0 and UOA_WB_1. **a** Venn diagram of structural differences called in UMD_CASPUR_WB_2.0 and haplotigs when UOA_WB_1 was used as the reference. The 8664 unique and 1313 overlapping differences in haplotigs represent heterozygous alleles. Structural differences present only in UMD_CASPUR_WB_2.0 are likely assembly errors. **b** Total bases of structural differences in categories deletion, insertion, repeat contraction, repeat expansion, tandem contraction, and tandem expansion. For example, for deletion, we report the number of bases found in UOA_WB_1 but missing in UMD_CASPUR_WB_2.0. **c** Count of structural differences in the categories from part b, partitioned by size

contained 9989 structural variants (SVs) for a total of 10.4 Mb (Supplementary Figure 4) and carried 2,826,343 SNPs.

The assembled sequence contains 3841 complete single-copy orthologs and only 40 duplicated orthologs for the 4104 mammalian BUSCO gene groups (Supplementary Figure 5). Although the presence of 93.6% BUSCO completeness score indicates that the current assembly is of high quality, we caution using this metric for assembly evaluation. The previous short-read-based water buffalo assembly had a BUSCO score of 93.0% despite having a highly fragmented genome. PacBio-based assemblies of zebra finch and hummingbird also reported that BUSCO scores that were little improved when compared with intermediate and short-read-based assemblies[12].

**Sequence contiguity assessments**. A metric to assess the quality of a genome assembly is the number of gaps that interrupt sequence contiguity. Compared with the human reference (GRCh38) and the goat reference (ARS1) (Fig. 4), UOA_WB_1 has fewer gaps and is more contiguous. Only the X chromosome, with 65 gaps, compared unfavorably with the human X chromosome (28 gaps). The human genome still has the longest un-gapped contig of 141.4 Mb (on chromosome 2). The longest un-gapped contig in the water buffalo genome is 104.7 Mb (on chromosome 1); whereas, the longest un-gapped goat contig is 87 Mb (on chromosome 11). Chromosome 24 of UOA_WB_1 is the most complete buffalo chromosome with only a single gap.

**Resolution of longer repeats**. The assembly strategy used for UOA_WB_1, based on long PacBio reads, substantially improved repeat resolution when compared with UMD_CASPUR_WB_2.0. Over 47.48% of the assembly consists of repeat elements, which is consistent with other published mammalian assemblies, including the human GRCh38 and the goat ARS1. The UOA_WB_1 buffalo assembly has a 1.59% higher repeat content than the UMD_-CASPUR_WB_2.0 assembly. A quarter of the genome is covered by two large repeat families, which are long interspersed nuclear element (LINE) L1 and LINE/RTE-BovB (Supplementary Figure 6). Scaffolds that could not be placed on chromosomes would be expected to be rich in repeats, and indeed 23% of the unplaced scaffolds are comprised of centromeric repeats. The next most abundant repeat types in unplaced scaffolds are LINE/L1 and LINE/RTE-BovB elements, which together account for another 16% of bases in unplaced scaffolds. The centromeric, LINE L1 and BovB repeat-rich regions account for most of the breaks in sequence contiguity. UOA_WB_1 has more repeats >2 kb when compared with the previous short-read based water buffalo assembly (Fig. 5a). Additionally, the LINE L1, BovB and centromeric repeats present in UOA_WB_1 are longer than those in the goat ARS1. Chromosomes 1–5 of the water buffalo are sub-metacentric, and centromeric repeats were found at the expected locations where homologous cattle chromosomes are joined together[18]. For example, water buffalo chromosome 1 (202 Mb) is homologous to cattle chromosome 27 (45 Mb) joined with cattle chromosome 1 (158 Mb) and centromeric repeats are found at the junction. A total of 15 out of 25 chromosomes have centromeric repeats >5 kb illustrating that UOA_WB_1 is a true chromosome-level assembly. Seven acrocentric autosomes have centromeric repeats within 100 kb from the chromosome ends, suggesting the assembly approaches the telomeres. However, the assembly of telomeres is difficult and searches for the ubiquitous vertebrate telomeric repeats (TTAGGG)n did not identify any chromosome with resolved telomeres.

**Improved gene annotation**. Annotation of UOA_WB_1 was carried out using ~15 billion RNA-Seq reads from over 50

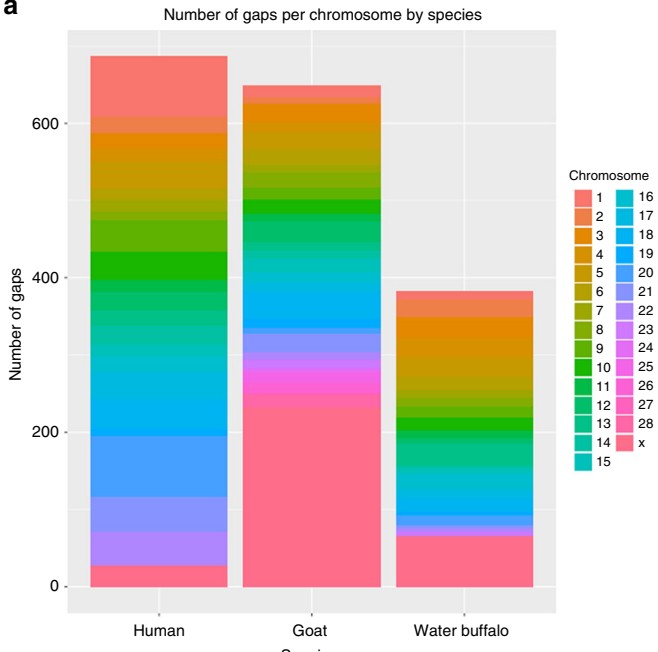

**a**
Number of gaps per chromosome by species

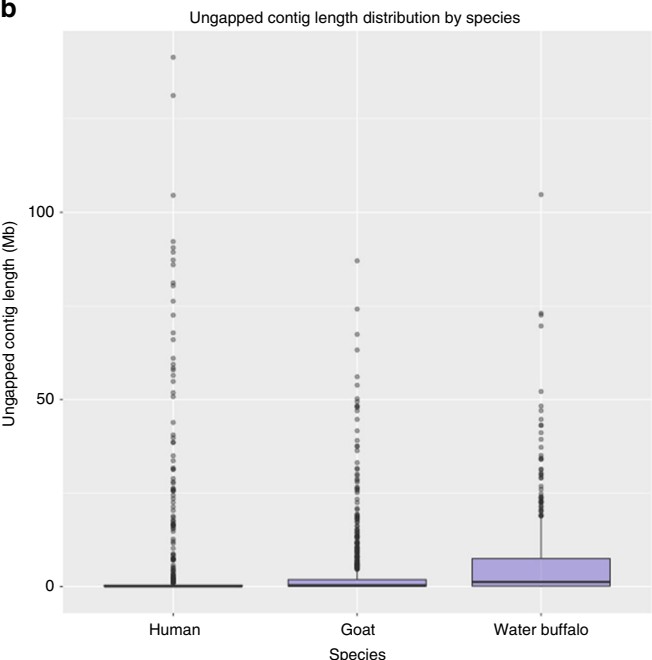

**b**
Ungapped contig length distribution by species

**Fig. 4** Comparisons of gaps and sequence contiguity between human, goat, and water buffalo assemblies. **a** Barplot of number of gaps by chromosomes. **b** Distribution of un-gapped contig lengths between the assemblies of the 3 species. Wilcoxon rank sum, one-sided test (water buffalo ($n = 480$) against human ($n = 687$), $W = 212,810$; water buffalo ($n = 480$) against goat ($n = 680$), $W = 165,300$; p-value after Bonferroni correction <0.05)

different tissues, which is ~10 times the quantity of RNA-Seq reads used to annotate UMD_CASPUR_WB_2.0 and more than those used to annotate the latest human genome GRCh38. A comparison of various assembly features between water buffalo, goat, and human genomes is given in Table 2. UOA_WB_1 contains a total of 20,801 protein-coding genes, 8443 non-coding genes, and 4465 pseudogenes. The full annotation report for the

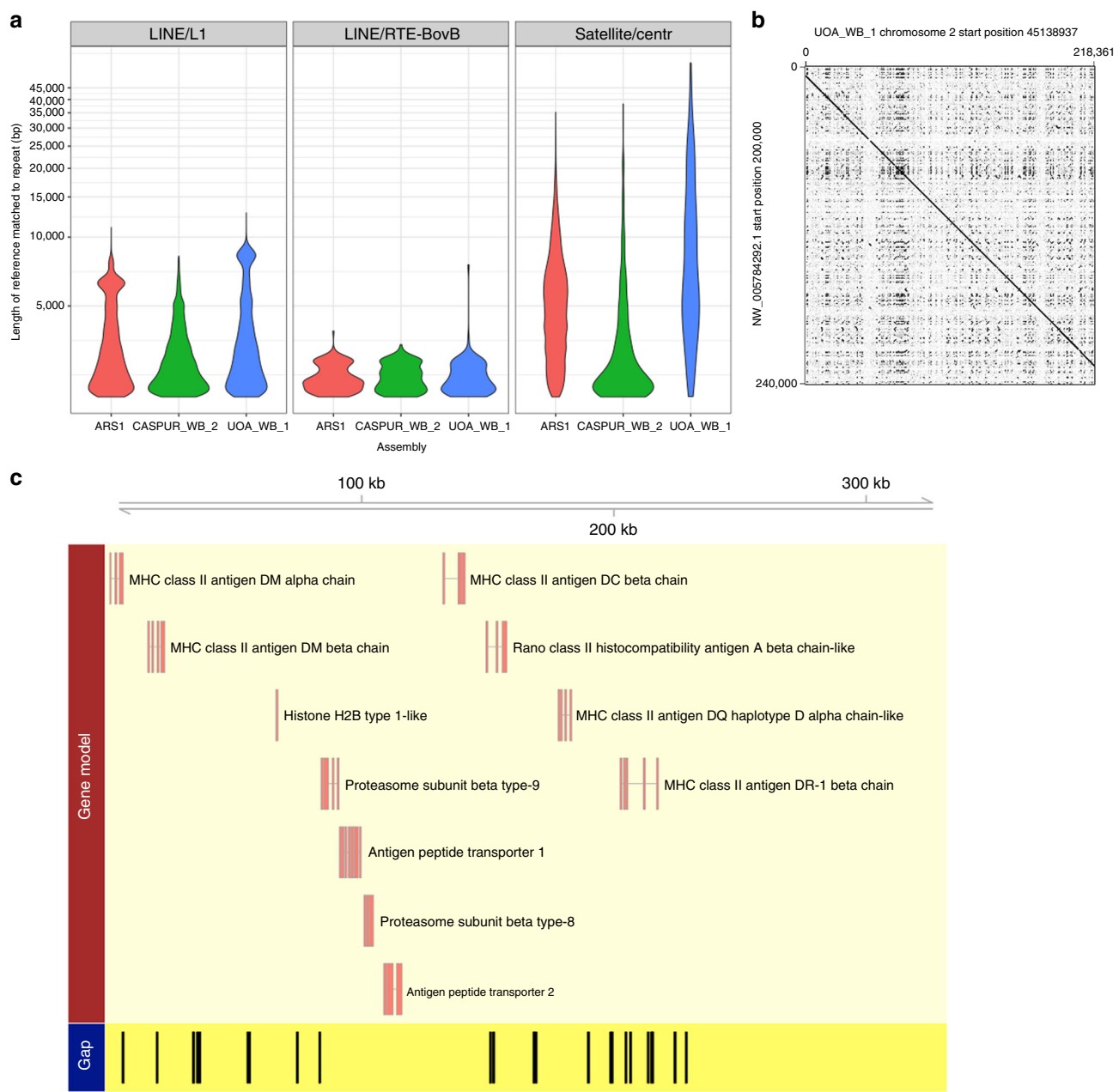

**Fig. 5** Resolution of hard to assemble repetitive and polymorphic regions. **a** Violin plot of repeat lengths >2 kb for LINE/L1, LINE/RTE-BovB and satellite/centromeric repeats for ARS1, UMD_CASPUR_WB_2.0 and UOA_WB_1 assemblies. **b** Dot plot of a ~218 kb region of MHC class II in UOA_WB_1 (horizontal) against UMD_CASPUR_WB_2.0 (vertical) showing a substantial level of repetition throughout the region. **c** Resolved MHC class II genes present on the single contig in UOA_WB_1 also shown in **b**. Protein-coding genes in UOA_WB_1 are shown for the same single contig, with assembly gaps for the same region in UMD_CASPUR_WB_2.0

current *B. bubalis* assembly is available in Annotation Release 101 (AR 101); whereas, the previous assembly is in Annotation Release 100 (AR 100) (see URLs). Only 3% of gene models are strictly identical between the current and previous assembly, 47% have undergone minor changes and 26% of annotated genes are considered novel, as no good match was found in the previous assembly. One indicator of the high quality of genome annotation is the presence of few partial coding sequence (CDS). UOA_WB_1 has only 157 partial CDS; ~10 times fewer than the previous assembly (Supplementary Table 4). The latest human annotation (GRCh38, NCBI Annotation Release 109) and goat

annotation (ARS1, NCBI Annotation Release 102) contain 533 and 457 partial CDS, respectively. Another indication that UOA_WB_1 is an improvement over UMD_CASPUR_WB_2.0 is the increase in the mean and median CDS length from 1787 bp and 1332 bp in AR 100 to 2031 bp and 1500 bp in AR 101, which are values similar to the latest human annotation. The percentage of CDSs with major correction in water buffalo (UOA_WB_1), in which a base insertion or deletion relative to the genomic sequence was introduced in order to maintain the frame of the protein is 9% and comparable to some recent PacBio-based reference assemblies also annotated by the NCBI Eukaryotic

**Table 2 Assembly features**

| Feature[a] | Water buffalo (UOA_WB_1) | Water buffalo (UMD_CASPUR_WB_2.0) | Goat (ARS1) | Human (GRCh38) |
|---|---|---|---|---|
| Genome size (Gbp) | 2.66 | 2.84 | 2.92 | 3.26 |
| Repeat (%)[b] | 47.48 | 45.89 | 50.58 | 49.95 |
| Genes count | 29,244 | 24,014 | 24,766 | 38,096 |
| Coding sequences count | 58,204 | 41,486 | 42,674 | 113,633 |
| Introns count | 238,481 | 209,659 | 209,898 | 351,892 |
| Exons count | 269,697 | 234,918 | 236,566 | 408,659 |
| Transcripts count | 71,537 | 47,030 | 48,672 | 160,474 |
| Mean number of transcripts per gene | 2.54 | 1.91 | 2.05 | 4.18 |
| Mean number of exons per transcript | 11.79 | 10.73 | 12.07 | 11.72 |

Comparisons of various assembly features of the water buffalo, goat, and human genomes as annotated by NCBI (water buffalo UOA_WB_1: NCBI AR 101, water buffalo UMD_CASPUR_WB_2.0: NCBI AR 100, goat ARS1: NCBI AR 102, human GRCh38.p12: AR 109)
[a]Pseudogenes are excluded
[b]Percentage of sequences masked by RepeatMasker

Genome Annotation Pipeline: swine (Sscrofa11.1, 11%), cat (Felis_catus_9.0, 8%), the Egyptian bat (Raegyp2.0, 11%), but higher than for goat (ARS1, 4%) and horse (EquCab3.0, 3%).

The high sequence contiguity of the current assembly allows the hard to assemble gene clusters to be resolved and annotated. As an illustration, the major histocompatibility complex (MHC) II region is fully assembled. The MHC plays a pivotal role in initiating immune responses and hence it is important for disease resistance[23]. The MHC is in a gene dense region and contains highly polymorphic loci and long-repetitive sequences. This structural complexity has made it extremely difficult to assemble the MHC region[24]. Without any additional information such as BAC sequencing, the MHC class II region was assembled as one contig, spanning ~218 kb whereas the equivalent region in UMD_CASPUR_WB_2.0 has 26 gaps (Fig. 5b, c).

## Discussion

The goal of a genome project is a finished haplotype-resolved assembly with no gaps. Closing gaps requires significant painstaking effort[3], and even with the availability of long reads, gaps are likely to remain open while filled gaps may contain errors[25]. No mammalian genome is completely assembled and gap free but it is now feasible to obtain near-finished haplotype-resolved assemblies using the methodology described here for the *B. bubalis*. Despite a degree of homozygosity in the animal sequenced, with the 75x PacBio coverage it was possible to assign 58% of the genome to haplotigs and to surpass the sequence contiguity of both the latest goat and the human reference genomes. This is partly because PacBio reads used in this assembly were on average 11.5 kb, more than twice the length of those used for the goat assembly[7]. Better sequence contiguity and ~58% of the genome phased led to improved gene annotation[12], which surpasses the goat genome annotation when using a count of partial CDS as the quality measure.

Nevertheless, even with the long sequence reads, contiguity is interrupted by repeats such as centromeres and LINEs, which necessitates the use of scaffolding technologies. The use of Chicago[11] and Hi-C[9] here achieved longer range scaffolding, approaching chromosome-level assembly. Other techniques including optical mapping from BioNano[26] may further improve the assembly quality, even though join accuracy is reported to be ~15% higher in Chicago[27]. Furthermore, the Chicago-based methods incorporate more smaller scaffolds (<100 kb) than optical mapping. After the initial PacBio FALCON-Unzip contig assembly, the median contigs length was 67,420 bp, which argues that Chicago is a better choice than an optical map. However, better results may come with the use of both Chicago and optical

mapping as the two technologies have different advantages and biases. The goat assembly, which used optical mapping but not Chicago, contains six autosomes with telomeric sequences whereas the water buffalo has none. The Chicago method relies on mapping short Illumina reads, which may miss the telomeric regions that are highly repetitive with $(TTAGGG)_n$.

Increased accessibility of short-read sequencing has resulted in a deluge of species with genome assemblies; mostly incomplete and fragmented. Using long-read PacBio sequencing we covered many regions missing from Illumina-based sequence from the same individual, and were able to assemble 19 regions each larger than 8 kb that were undetected in the short-read data. A major advantage of long-read sequencing is the inclusion of large repeat families, such as LINE L1 and BovB that are not properly assembled by short-read-based methods. In the absence of this information evolution of these elements which differ among species and may influence gene expression (e.g. 16% of genome *B. bubalis* is made up of LINE L1 and BovB) cannot be studied.

The HiRise and FALCON-Unzip software sometimes gave conflicting information, mainly in regions where there are haplotype phase switches. Genome sequences generated by early adopters of the FALCON-Unzip and HiRise (e.g. durian genome[28]) may therefore contain false contig breaks. We have created custom scripts to rejoin such false breaks but in the future assemblers such as FALCON-Phase[16] that integrates Chicago/Hi-C data directly may better deal with this problem. Besides haplotype phase switches, the breaks identified by HiRise around regions with high coverage indicate potential segmental duplication that might be tandem or interspersed. In the case of tandem duplication, the assembly may have compressed such repeats leading to a higher than expected coverage and hence, a break to the contig is appropriate. If the high coverage region results from interspersed segmental duplication and the contig is indeed correct, breaking it should not be a problem because the gap filling step should refill the gap.

The water buffalo assembly reported here demonstrates that the combination of long-read sequencing with serial Chicago and then Hi-C scaffolding produces a very high-quality chromosome-level mammalian genome assembly. Additional information used included the LD mapping and conservation of synteny with the cattle and goat genomes, to refine and validate the assembly, but these did not lead to substantial improvements. Additionally, we used short paired-end reads to correct indels, but found that only ~0.014% of the genome or 0.37 Mb required correction (Supplementary Table 5). As long-read sequencing chemistry continues to improve, the use of short reads for assembly polishing may become unnecessary. Long-read sequencing coupled with

chromatin conformation capture technologies is currently one of the best approaches to generate high quality genome assemblies without the need for a pre-existing reference.

## Methods

**Chosen animal.** A female Italian Mediterranean buffalo, Olimpia, the offspring of a half-sib mating previously used for a draft genome assembly based on short reads[19] (GenBank assembly accession: GCA_000471725.1) was chosen for sequencing. Olimpia has a normal river buffalo karyotype ($n = 25$; $2n = 50$) as verified by high resolution R-banding[19]. Blood samples were collected for sequencing. All animal work was done in compliance with Italian laws on animal experimentation and ethics (DL n. 116, 27/01/1992).

**Genome sequencing and assembly of contigs.** Seven libraries for SMRT sequencing were constructed from blood derived genomic DNA, using SMRTbell Template Prep Kit v1.0 (Pacific Biosciences, Menlo Park, CA; "PacBio"). After library construction, size selection was performed on a BluePippin instrument (Sage Science, Beverley, MA) with size cutoff set at 30 kilobases (kb). A total of 8 SMRT cells were run on the RSII instrument (PacBio) using the P6/C4 chemistry, to test each library prior to production runs totaling 57 SMRT Cell v1M on the Sequel instrument (PacBio) using Sequel Sequencing Kit v1.2 chemistry. A total of 199.2 Gbp was generated with mean read length of 5.8 kb for RS II data and 11.5 kb for Sequel data, respectively: 96% of the sequence yield that comprises 191 Gb of data came from the Sequel platform. Assuming a genome size of 2.65 Gbp, the raw PacBio data represent ~×75 coverage.

The de novo assembly of contigs was performed with FALCON[15] version 0.7.0 and FALCON-Unzip (see URLs). Briefly, reads longer than 5 kb were selected as "seed" reads for error correction ("preassembly"). Preassembly in FALCON uses DALigner to do all-by-all alignments of the raw reads. The use of sensitive DALigner parameters (-k14 -h256 -l1200 versus -k18 -h1250 -l1500) resulted in a higher pre-assembled yield; measured as the total length of pre-assembled reads divided by the total length of seed reads. See Supplementary Note 1 for the configuration file. The FALCON assembly resulted in 1694 primary contigs of total length 2.66 Gb, contig N50 of 18.7 Mb and an additional 0.22 Gb of "associate contigs" that represent divergent haplotypes in the genome. The FALCON-Unzip module was then applied, whereby raw reads are phased according to SNPs identified in the draft FALCON assembly and then reassembled in separate haplotype-specific pools. FALCON-Unzip produces contiguous primary contigs and more fragmented haplotigs, which represent phased, alternate haplotypes. The genome assembly was polished twice: first as part of the FALCON-Unzip pipeline using haplotype-phased reads, and then second, using the "resequencing" analysis application of SMRT-Link v4.0.0 with default parameters and primary contigs and haplotigs concatenated into the single reference. In resequencing, all reads were aligned to the genome assembly contigs using BLASR and then consensus sequences were called using the arrow algorithm. The final FALCON-Unzip assembly had 953 primary contigs and 7956 haplotigs.

**Chicago library preparation and sequencing.** Three Chicago libraries were prepared as described previously[11]. Briefly, for each library, ~500 ng of genomic DNA (mean fragment length of 75 bp) was reconstituted into chromatin in vitro and fixed with formaldehyde. Fixed chromatin was digested with DpnII, the 5' overhangs filled in with biotinylated nucleotides, and free blunt ends ligated. After ligation, crosslinks were reversed and the DNA purified. Purified DNA was treated to remove biotin that was not internal to ligated fragments. The DNA was then sheared to ~350 bp mean fragment size and sequencing libraries were generated using NEBNext Ultra enzymes and Illumina-compatible adapters. Biotin-containing fragments were isolated using streptavidin beads before PCR enrichment of each library. The libraries were sequenced on an Illumina NextSeq500. The number and length of read pairs produced for each library was: 87 million, $2 \times 151$ bp for library 1; 55 million, $2 \times 151$ bp for library 2; 67 million, $2 \times 151$ bp for library 3. Together, these Chicago library reads provided ×95 physical coverage of the genome (1–100 kb pairs).

**Dovetail Hi-C library preparation and sequencing.** Three Dovetail Hi-C libraries were prepared as described previously[9]. Briefly, for each library, chromatin was fixed in the intact nucleus with formaldehyde. Fixed chromatin was processed in the same way as for the Chicago library preparation. The libraries were sequenced on an Illumina HiSeq X (rapid run mode). The number and length of read pairs produced for each library was: 169 million, $2 \times 151$ bp for library 1; 176 million, $2 \times 151$ bp for library 2; 168 million, $2 \times 151$ bp for library 3. Together, these Dovetail Hi-C library reads provided ×5191 physical coverage of the genome (10–10,000 kb pairs).

**Scaffolding with HiRise.** The 953 primary contigs from the FALCON-Unzip assembly and Chicago reads were used as inputs for the Dovetail HiRise Scaffolding software[11]. The program is specifically designed to use proximity-ligation data to scaffold contigs. Briefly, the process starts by aligning Chicago reads to the primary

contigs assembly using a modified SNAP aligner[29] (https://github.com/robertDT/dt-snap) with parameters "-ku -as -C-+-tj GATCGATC -mrl 20". A likelihood model is then built based on the mapping distances of read pairs. The scaffolding process makes decisions on contig breaks and joins iteratively to arrive at an assembly that best fits the model.

The primary contigs were broken at 108 positions and 293 joins were made. The large number of breaks introduced to the primary contigs suggested that some of the breaks were incorrect. Breaks created were therefore tested as follows. For each break, a 50 kb window with the breakpoint at the center was assessed for the PacBio sequence coverage and Chicago read pair distance. In some cases incompatibilities in the use of primary contigs as input assembly for HiRise scaffolding were identified. These errors occurred mainly where there was a phase switch in the FALCON-Unzip assembly. Custom scripts were written to identify false breaks, which were identified as a HiRise breaks where the PacBio sequence coverage was normal. Contigs were only joined based on high confidence breaks and joins. After scaffolding and error correction with Chicago reads the resulting scaffolds were used as input for a second round of HiRise scaffolding using Hi-C reads. The same methods were used to explore and confirm breaks and joins in scaffolds. The clustering of scaffolds into a chromosome-scale assembly is given in Supplementary Figure 7.

**Checking scaffold joins.** Currently 388 loci are mapped on the cytogenetic map[30] and 3093 loci are present on the radiation hybrid (RH) map[18] for the water buffalo. The limited number of loci physically mapped provided insufficient resolution to confirm the precise order and orientation of scaffolds. Instead we used linkage disequilibrium (LD) data coupled with conservation of synteny between buffalo with cattle and goat genomes to validate the assembly, order and orientation of contigs in scaffolds. A linkage disequilibrium (LD) map for the buffalo was created using the LDMAP program[31] from SNP genotype data. Briefly, the genotype data came from 529 animals assayed on the 90 K buffalo Axiom chip[32]. First, the SNP sequences were mapped to the new reference using blastn. To test for scaffolds that might belong together, each scaffold was joined to all other scaffolds in all possible orientations and these synthetic joins were checked for changes in LD that would be consistent with them being contiguous. Similarly, scaffolds were analyzed for internal jumps in LD that would be consistent with underlying contigs not being correctly assembled (Supplementary Figure 8). The low density of the SNP data meant that only major scaffolds carried sufficient SNPs to be tested in this manner. For each SNP, LDMAP gives a location in LD units[33] (LDUs) and intervals between apparently adjacent SNPs which span a large LDU distance suggest weak LD across the interval. These larger LD jumps are indicative of potential mis-assembly. Altogether, 58,588 LD jumps were identified and the outlier threshold value based on standard scaffolds was 0.275 (Supplementary Figure 8). Any region with LD jump higher than the outlier threshold was treated as a potential mis-join.

After scaffolds were built with serial Chicago and Hi-C assembly, the scaffolds contained 484 gaps. Each gap was the join of two contigs. To check for conservation of synteny, the left and right 3 kb sequences of each gap were used as input for blastn searches against the UMD3.1 bovine[34] and ARS1 goat[7] genomes. The blastn parameters were set to keep alignments with e-value less than $1e^{-10}$ and percent identity more than 85%. A gap was defined as having conserved synteny if the left and right sequences had blast hits to the same target chromosome, same strand, had an alignment length of 1 kb or more and were within 1 Mb of each other.

**Gap filling and polishing.** After checking scaffolds with LD data and conserved synteny, the scaffolds that contain 488 gaps were gap filled with PBJelly[35] v15.8.24 using all raw PacBio Sequel subreads. PBJelly was run with default parameters except for the support module, where the options "captureOnly and spanOnly" were used. This step closed 54 gaps that further add support to the contig joins surrounding these gaps. A final round of BLASR and arrow (see URLs) was run to polish the scaffolds to give quality scores to gap filled sequences. Finally, an additional ~×80 coverage of paired-end Illumina WGS library was generated for sequence polishing using Pilon v1.22[36]. The insert size for the Illumina library was 350 bp and sequencing was on a NextSeq500 generating $2 \times 150$ bp reads using a 300 cycle kit with 1% PhiX spike-in. Illumina reads were aligned to the polished gap filled assembly using BWA v0.7.12[37] and SAMtools v0.1.18[38]. Pilon was run with the parameters "–diploid –fix indels –nostrays" to correct the insertion/deletion errors that are more common in PacBio reads. There were approximately 3.5 times more insertions (145,105 events) than deletions (41,409 events) corrected with Pilon (Supplementary Table 5).

The final assembly passed the NCBI foreign contamination screens that filter out common contaminants such as vectors, bacterial insertion sequences, E. coli, phage genomes, adaptor linkers and primers, mitochondria, chromosome of unrelated organisms and ribosomal RNA genes.

**Assembly evaluation.** The completeness of the genome from contig to chromosome-level assembly was assessed using the benchmarking universal single-copy orthologs (BUSCO) v2.0.1[39]. The mammalia_odb9 lineage-specific profile that contains 4104 BUSCO gene groups was tested against assemblies of the water

buffalo using the option "-m geno". (Supplementary Figure 6). Further detail on assembly evaluation is given in Supplementary Note 1.

**Genome annotation**. The NCBI Eukaryotic Genome Annotation Pipeline was used to annotate genes, transcripts, proteins and other genomic features[40]. The methodology for annotation is as described for the UMD_CASPUR_WB_2.0 assembly[19]. The evidence used as input for this annotation run included 3462 buffalo transcripts present in Genbank and dbEST, 1013 buffalo Genbank protein sequences, 50,553 human RefSeq proteins (with NP_ prefix), 13,381 *Bos taurus* known RefSeq proteins and 15.6 billion RNA-Seq reads from over 50 different buffalo tissues.

**Repeats analysis**. RepeatMasker version open-4.0.6 (see URLs) was used to search for repeats in the current assembly by identifying matches to RepBase[41] and RepeatMasker database both version 20150807. Results of repeat searches of the previous short-read water buffalo assembly (GCF_000471725) and goat assembly (GCF_001704415.1) were downloaded from the NCBI. Only repeats with matches above 60% identity were used for analysis. Centromeric repeat analysis was carried out using the cattle and sheep repeats that belong to the family 'Satellite/centr' within Repbase. RepeatMasker by default searches for 6-mer TTAGGG, which is the vertebrate telomeric repeat. Chromosome ends defined as within 100 kb of sequence ends were searched for telomeric repeats.

**Gap comparisons and sequence contiguity**. Two of the best mammalian genome assemblies, the human genome assembly (GRCh38.p12) and goat assembly (ARS1), were downloaded from the NCBI for the evaluation of gaps and sequence contiguity against the buffalo genome. Only sequences that belong to autosomes and X chromosome were retained for analysis, whereas unplaced, unlocalised, mitochondrial and Y chromosome sequences were filtered out. The tool seqtk v1.2-r94 (see URLs) was used to generate positions of gaps with minimum of three Ns, as well as un-gapped contigs that result from breaking of scaffolds at each gap position (Supplementary Note 1). Using this method, the 649 gaps reported in the goat genome[7] were reproduced. The number of gaps and un-gapped contig length distribution were analysed using custom R scripts.

**Statistical analysis**. R/Bioconductor was used for all statistical analyses. Wilcoxon rank-sum test with continuity correction was used to compare un-gapped contigs of human, goat and water buffalo using the function wilcox.test for a one-sided test of whether the buffalo has longer sequence contiguity at $P < 0.05$ after Bonferroni correction for multiple tests.

**Code availability**. Custom scripts can be found at GitHub repository at the following URL: (https://github.com/lloydlow/BuffaloAssemblyScripts)

**URLs**. Arrow, https://github.com/PacificBiosciences/GenomicConsensus; seqtk, https://github.com/lh3/seqtk; FALCON-Unzip, https://github.com/PacificBiosciences/FALCON_unzip; RepeatMasker, http://www.repeatmasker.org; Annotation Release 101, https://www.ncbi.nlm.nih.gov/genome/annotation_euk/Bubalus_bubalis/101/; Annotation Release 100, https://www.ncbi.nlm.nih.gov/genome/annotation_euk/Bubalus_bubalis/100/

**Reporting summary**. Further information on experimental design is available in the Nature Research Reporting Summary linked to this article.

## Data availability

The PacBio reads, Chicago reads, Hi-C reads, and Illumina paired-end reads are available in SRA under BioProject PRJNA437177. The RNA-seq reads can be obtained from BioProject PRJEB25226 and PRJEB4351. The previous short-reads-based water buffalo assembly, GCF_000471725.1, was downloaded from the NCBI. Intermediary assembly FASTA files and other miscellaneous information are available from the corresponding authors upon request.

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

## Acknowledgements

Special thanks to Alessio Valentini for the initial discussion on using LD to check the assembly. J.L.W., R.T., and W.Y.L. are supported by the JS Davies Bequest to the University of Adelaide. This work was supported in part by the intramural research program of the National Library of Medicine, National Institutes of Health (F.T.-N., T.M.). D.M.B. and B.D.R. were supported by USDA CRIS project number 8042-31000-001-00-D. D.M.B. was also supported by USDA CRIS project number 5090-31000-026-00-D. T.P.L.S. was supported by USDA CRIS project number 3040-31000-100-00-D. This research used resources provided by the SCINet project of the USDA Agricultural Research Service, ARS project number 0500-00093-001-00-D. Mention of trade names or commercial products in this article is solely for the purpose of providing specific information and does not imply recommendation or endorsement by the US Department of Agriculture.

## Author contributions

J.L.W. and T.P.L.S. conceived and managed the project; W.Y.L. analysed all data from raw PacBio reads contig assembly to final genome submission to the NCBI; R.T. and A.C. performed the LD analysis and water buffalo alignment with cattle; D.M.B. evaluated the genome assembly; D.M.B. and B.D.R. provided guidance on assembly analysis; S.B.K. performed FALCON-Unzip assembly and genome polishing with PacBio reads; T.S. performed scaffolding with HiRise; P.A.M. provided sample of Olimpia for sequencing; F.T.-N. and T.M. annotated the assembled genome; D.H., R.Y., and L.L. provided access to RNA-Seq data for genome annotation; W.Y.L. and J.L.W. drafted the manuscript and all authors read, edited, and approved the final manuscript.

## Additional information

**Competing interests:** S.B.K. is an employee of Pacific Biosciences, T.S. is an employee of Dovetail Genomics.

