## [Peer Review File · Nature Communications]

Reviewer #1 (Remarks to the Author):

Low et al. describe the assembly of a water buffalo genome using a combination of PacBio, Chicago, and Hi-C sequencing data followed by corrections using linkage disequilibrium maps. Gap fill is then performed using PBJelly followed by Illumina error correction using Pilon. The chromosomes were then scaffolded using radiation hybrid maps and cow synteny. The quality of the assembly is assessed by comparisons to a previous Illumina assembly of the same animal including comparisons of NCBI gene annotations. Other quality assessments are performed such as running BUSCO and comparing assembly statistics like number of gaps to human and goat assemblies. There are also results presented for resolution of LINE repeats and presence of centromeric repeats.

The paper is well written and the technical aspects of performing the assembly and quality assessment are good. The assembly process uses fairly standard tools such as FALCON-Unzip and HiRise and does not present novel assembly methods. However, the process is well described and could assist other researchers who are pursuing a similar assembly strategy. The caveat about HiRise generating potentially false breaks is useful. There is little in the way of biological insights gained from the paper since genome assembly is the focus. However, the new water buffalo genome assembly is a great improvement over the previous short read assembly and should prove to be a valuable resource for studies of water buffalo and other agriculturally relevant bovine species.

Major Issues

None

Minor Grammatical Issues

P4 - Chicago, a modified form of Hi-C uses chromatin...

Should have a comma after Hi-C and read

Chicago, a modified form of Hi-C, uses chromatin...

It looks like the authors addressed most of reviewer 1's comments. The only suggestion I would have is in regards to this new sentence added based on comment 1:

"Olimpia has a normal river buffalo karyotype with 50 haploid chromosomes as verified by high resolution R-banding."

It sounds slightly odd since 50 is the diploid number, but they say haploid chromosomes. Maybe something like this instead:

"Olimpia has a normal river buffalo karyotype ($n=25$; $2n=50$) as verified by high resolution R-banding."